# Stable Oxygen and Carbon Isotope Composition of Holocene Mytilidae from the Camarones Coast (Chubut, Argentina): Palaeoceanographic Implications

**Gabriella Boretto [1], Giovanni Zanchetta [2,3,4,*], Ilaria Consoloni [2], Ilaria Baneschi [5]**, **Massimo Guidi [5], Ilaria Isola [6,7], Monica Bini [2,3], Luca Ragaini [2], Filippo Terrasi [8]**, **Eleonora Regattieri [5] and Luigi Dallai [5,6,9]**

[1] Universidad Nacional de Córdoba. Facultad de Ciencias Exactas, Físicas y Naturales. Consejo Nacional de Investigaciones Científicas y Técnicas (CONICET). Centro de Investigaciones en Ciencias de la Tierra (CICTERRA). Av. Vélez Sarsfield 1611, X5016GCA Córdoba, Argentina; gmboretto@yahoo.com.ar

[2] Dipartimento di Scienze della Terra, Università di Pisa, Via S. Maria 53, 56126 Pisa, Italy; i.consoloni@gmail.com (I.C.); monica.bini@unipi.it (M.B.); luca.ragaini@unipi.it (L.R.)

[3] CIRSEC Centre For Climatic Change Impact, University of Pisa, Via del Borghetto 80, 56124 Pisa, Italy

[4] Istituto di Geologia Ambientale e Geoingegneria, IGAG-CNR, Via Salaria km 29,300, 00015 Montelibretti, Rome, Italy

[5] Istituto di Geoscienze e Georisorse IGG-CNR, Via Moruzzi 1, 56100 Pisa, Italy; i.baneschi@igg.cnr.it (I.B.); guidimassimo1@gmail.com (M.G.); eleonora.regattieri@igg.cnr.it (E.R.); dallai.luigi@gmail.com (L.D.)

[6] Istituto Nazione di geofisica e Vulcanologia, Via della Faggiola 32, 56100 Pisa, Italy; ilaria.isola@ingv.it

[7] Istituto Nazione di geofisica e Vulcanologia, Via Vigna Murata 605, 00143 Roma, Italy

[8] CIRCE, Department of Environmental Sciences, Second University of Naples, Viale Lincoln, 5, 81100 Caserta, Italy; filippo.terrasi@unicampania.it

[9] Dipartimento di Scienze della Terra, Università di Roma "La Sapienza", Ple A. Moro 5, 00185 Roma, Italy

**\*** Correspondence: giovanni.zanchetta@unipi.it

**Abstract:** The stable isotope composition of living and of Holocene Mytilidae shells was measured in the area of Camarones (Chubut, Argentina). The most striking results were the high $\delta^{18}O$ values measured in samples older than ca. 6.1 cal ka BP. In the younger samples, the $\delta^{18}O$ values remained substantially stable and similar to those of living specimens. Analysis of the data revealed the possibility for this isotopic shift to be driven mainly by changes in temperature probably accompanied by minor changes in salinity, suggesting cooler seawater before 6.1 cal ka BP, with a maximum possible temperature shift of ca. 5 °C. A possible explanation of this change can be related to a northward position of the confluence zone of the Falkland and Brazilian currents. This is consistent with the data obtained in marine cores, which indicate a northerly position of the confluence in the first half of the Holocene. Our data are also in line with the changes in wind strength and position of the Southern Westerlies Wind, as reconstructed in terrestrial proxies from the Southernmost Patagonia region.

**Keywords:** Patagonian coast; mollusk shells; Falkland (Malvinas) current; Brazilian current; raised beaches; Holocene; stable isotopes

## 1. Introduction

The Brazilian Current flowing southward and the Falkland (Malvinas) Current flowing northward dominate the upper-level circulation of the Atlantic coast of Southern South-America (Figure 1a). The Falkland Current is a branch of the Circumpolar Current and flows along the continental shelf of

Argentina until it reaches the Brazil Current offshore the Río de la Plata estuary at ca. 38° [1,2]. Proxy data show that the front position has changed at millennial and at glacial/interglacial scales [3,4], as a result of changes in the position of the South Westerly Winds and of the long-term variations in the sea-surface temperature (SST) anomaly in the Agulhas-Benguela current [4,5]. At the inter-hemispheric scale, the position of the confluence of the Brazilian and of the Falkland currents also seems to be related to variations in the Atlantic Meridional Overturning circulation (AMOC) strength [4]. The changes in the front position have probably affected the biogeographic distribution of marine mollusks in the southern coast of South America. For instance, during the Middle Holocene, the presence of abundant mollusc fauna of 'Brazilian affinity' along the coast of the Bonaerense region, in a southern position compared to their current habitat, has been interpreted as a further penetration southward of the Brazilian Current [6]. Similarly, coastal marine successions over the northern Patagonia contain some differences in the marine fossil association between interglacials, which may be linked to changes in the position and strength of the Falkland and Brazilian currents [7–11]. These have pronounced differences in terms of temperature, salinity and oxygen isotope composition, with the Falkland current characterised by colder and fresher conditions compared to the Brazilian one [12,13]. The changes in their relative position could be identified by using the oxygen isotope composition ($\delta^{18}$O) of biological carbonates. For instance, [14] found that there is currently a strong oxygen isotopic gradient at the confluence of the Brazilian and Falkland currents (>2‰) in the calcite of *Globorotalia inflata* and *G. truncatulinoides*, collected at the top of the marine cores. This gradient was used by [4] to characterise the shift in the position of the confluence zone during the Holocene. Attempts have also been made to obtain palaeoceanographic information from coastal deposits outcropping along the Atlantic coast of Argentina by means of stable isotopes. Such efforts have been strongly influenced by the local mixture between continental and marine waters, overprinting possible isotopic effects related to regional changes in the oceanic circulation [15–21]. Recent data from Patagonia have been focussed on isotopic sclerochronology [22,23], which gives information on annual variability, whereas no clear interpretation has emerged from the study of bulk samples [23].

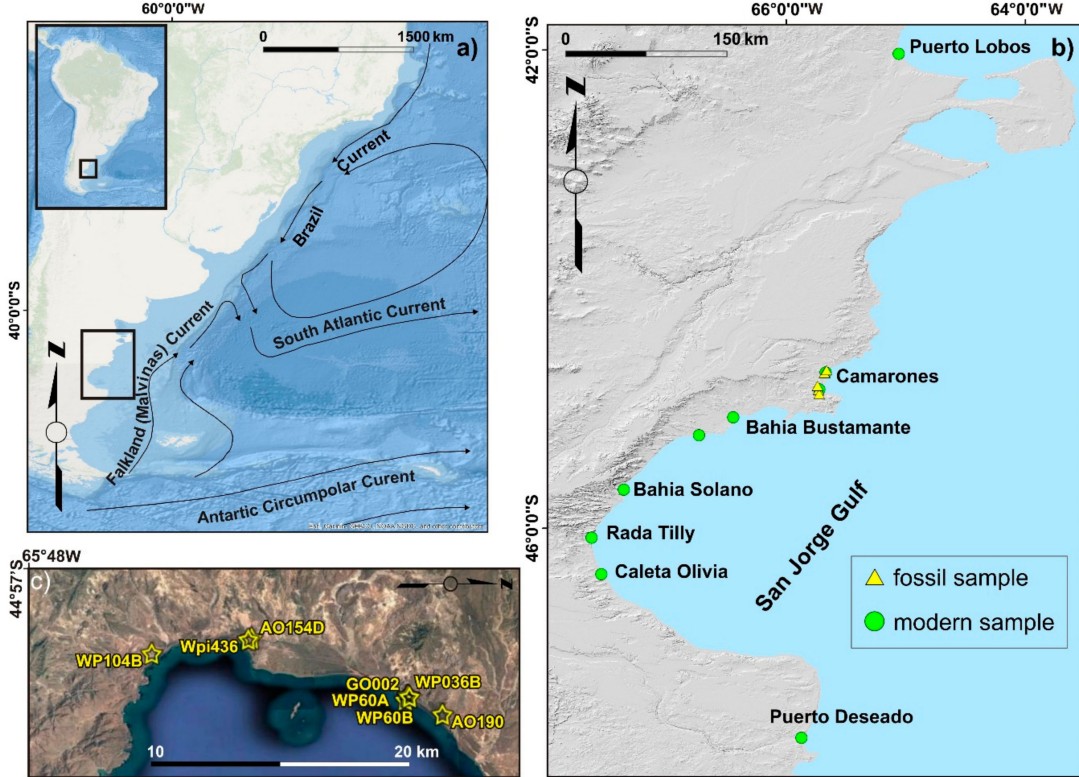

**Figure 1.** (**a**) General location map, with the position of the Falkland and Brazilian currents; (**b**) position of living and fossil shells collected in the study area; (**c**) Holocene sections sampled along the Camarones coast projected on Quick bird image. Labels correspond to those in Table 1.

In this paper, we discuss the stable oxygen and carbon isotope geochemistry of living and of Holocene shells of Mytilidae collected in sections situated near the Camarones village (Chubut, Argentina; Figure 1a–c). Our aim was to obtain information of possible palaeoceanographic changes on the coastal area of this sector of Patagonia since the Holocene transgression on land was recorded. The absence of significant river inflow reduced the possibility of water mixing effects and complications related to changes in local salinity.

## 2. Site Description

The study area is part of the Bahía Camarones, an approximately 40 km-wide gulf extending from ca. 44°54′ S to 44°34′ S (Figure 1b). Structurally speaking, the area is located on the southern edge of the so-called 'North Patagonia Massif'. Mainly Jurassic volcanic rock forms the pre-Quaternary succession [24]. Inland morphology is dominated by flat surfaces covered with gravely fluvial deposits ('*Rodados Patagonicos*', [25]) of poorly constrained age. Close to the coast, alternations of Quaternary littoral and continental deposits are present [24]. Quaternary marine coastal deposits have been extensively studied and mapped by [26–32]. Most of the recent studies on the Holocene transgression are referred to the southern part of the Bahía Camarones thanks to the intensive field work of Schellmann and Radtke [28–30], which supplied robust chronological constraints for coastal aggradation during the Holocene by using the radiocarbon dating method. The stratigraphy and chronology of these successions have been recently implemented by the works of [31–33]. Aguirre [7] and Aguirre et al. [10] provided a detailed description of the paleontological associations for different littoral units, including the Holocene sections.

The meteorological station of the Camarones village records a mean annual precipitation of 287 mm/a with a mean annual temperature of 12.6 °C. Throughout the year, the strong activity of the southern westerlies causes elevated soil evapotranspiration, turning the area into a semiarid and poorly vegetated land [34]. The potential evapotranspiration, calculated by the Thornthwaite method,

indicates an annual value of <700 mm, with a maximum of ≥100 mm in January (warm season) and a minimum of 15 mm in June (cold season). The annual evapotranspiration exceeds the rainfall, so the annual water deficit reaches approximately 450 mm, without considering the additional effect of the strong winds (http://www.ambiente.chubut.gov.ar).

Like most of coastal Patagonia, the area is dominated by high-energy, macro-tidal (>4 m) and stormy conditions [35], resulting in a coastal morphology dominated by cliffs, wave-cut platforms and coarse-clastic beach ridges ('swash built ridges' *sensu* [36]).

For this area, Falabella et al. [37] and the Argentine Naval Hydrography Service (http://www.hidro.gov.ar/ceado/Ef/Inventar.asp) reported an annual mean sea-surface temperature (SST) of around 12 °C; an average SST of 15.6 °C in the warm season; an average of 7.7 °C in the cold season; and a sea surface salinity (SSS) between ca. 33 and 33.5‰. This is in agreement with the satellite data, which indicate an average temperature of the sea off Camarones of 12 ± 3 °C for 2015 (http://seatemperature.org).

## 3. Materials and Methods

Several field campaigns were performed by the authors between 2009 and 2012. The samples were stored in the Laboratory of Paleoclimatology and Geoarchaeology of the University of Pisa. During the field campaigns, modern shells accumulated on the beach after sea storms were collected in several locations of S. Jorge's Gulf (Figures 1b and 2B).

Only shells with still-connected valves preserving the hinge ligament and with a fresh aspect, indicating that they had died recently, were selected. Whole shells were immersed for ca. 24 in a $H_2O_2$ solution in order to remove remains of organic matter. They were then washed in an ultrasonic bath and rinsed several times with deionised water. One shell of the pair was grounded in an agate mortar and was used for stable isotope analyses. A similar procedure was followed for fossil shells selected from Holocene deposits (Figure 1c). The powder obtained was not subject to further pre-treatments, to avoid undesirable isotopic effects.

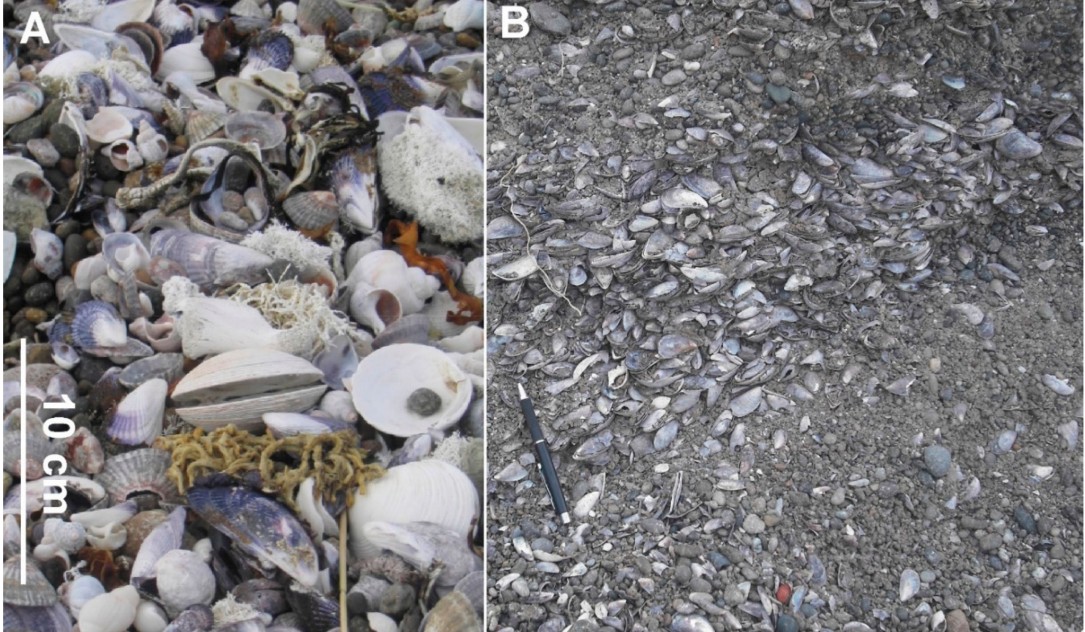

**Figure 2.** (**A**) Examples of shell accumulation over the Patagonian beaches; (**B**) examples of fossil storm accumulation.

The deposits in which the shells were collected are described in [30,31], which includes some radiocarbon dating (Table 1). Additional samples were collected and dated for this work (Table 1).

Similarly to modern specimens, the fossil shells were collected in deposits interpreted as storm accumulations (Figure 2B [31]).

In one case, the shell samples were collected from a small shellmidden [31]. Whole shells for radiocarbon dating were cleaned in an ultrasonic bath with the addition of $H_2O_2$ and then gently etched with diluted HCl. The radiocarbon measurements were performed at the CIRCE laboratory in Caserta [38,39]. Calibration was performed using the Marine13 curve in CALIB 6 [40], even if we need to take into account that the marine reservoir effect for the Patagonia marine shallow waters is poorly constrained and likely to be highly variable [30,41]. Therefore, the calibration offered by CALIB may be locally underestimated.

**Table 1.** Radiocarbon dating.

| Sample | Field Code | $^{14}$C a BP ($\pm 2\sigma$) | $^{14}$C cal a BP ($\pm 2\sigma$) (Median Probability) | Species |
|---|---|---|---|---|
| UCI65211 | WP60B * | 390 ± 50 | - | Aulacomya atra |
| UCI65210 | WP60A ** | 915 ± 20 | 477–560 (518) | *Aulacomya atra* |
| DSH2167 | G002 ** [1] | 4070 ± 50 | 3948–4271 (4106) | *Nacella (Patinigera) deaurata* |
| DSH2738 | AO154D | 5132 ± 67 | 5313–5608 (5493) | *Mytilus edulis* |
| DSH2745 | WPi436 | 5562 ± 43 | 5861–6092 (5949) | *Mytilus edulis* |
| DSH2734 | WP104B | 5675 ± 45 | 5947–6196 (6080) | *Mytilus edulis* |
| UCI65213 | WP63B ** | 6365 ± 20 | 6750–6913 (6834) | Mytilus edulis |
| DSH4026 | AO190 | 6486 ± 46 | 6867–7138 (6990) | *Mytilus edulis* |

Calibration following Reimer et al. [40]. * Modern specimen; ** After Zanchetta et al. [31]; [1] Shellmidden.

Carbonate powders were reacted under vacuum with 105% phosphoric acid at 70 °C and the evolved $CO_2$ gas was purified by using successions of cryogenic traps and then analysed with a MAT252 mass spectrometer available at IGG-CNR of Pisa. The isotopic results are reported by the conventional δ-notation in per mille (‰) values, namely percentage per thousand, and normalised to the Vienna Pee Dee Belemnite scale (V-PDB) using the internal working standards of Carrara Marble (MAB1), cross-checked against the NBS18 and NBS19 international reference materials. Mean analytical precision for both $\delta^{18}$O and $\delta^{13}$C is usually better than 0.15‰. The modern samples pertain to the Mytilidae family: *Aulacomya atra* and *Mytilus edulis* (Table 2).

**Table 2.** Stable isotope composition (oxygen and carbon) of modern samples collected along the S. Jorge Gulf (see Figure 1b for position).

| Locality | $\delta^{13}$C‰ (V-PDB) | $\delta^{18}$O‰ (V-PDB) |
|:---:|:---:|:---:|
| Camarones North | | |
| *Aulacomya atra* | | |
| | 1.8 | 1.4 |
| | 1.9 | 1.0 |
| | 2.0 | 1.3 |
| | 1.4 | 1.3 |
| | 2.0 | 1.2 |
| | 1.3 | 1.4 |
| Mean | 1.7 ± 0.3 | 1.3 ± 0.2 |
| Camarones South | | |
| *Aulacomya atra* | 1.1 | 1.4 |
| | 1.4 | 1.5 |
| | 1.6 | 1.3 |
| | 0.5 | 1.0 |
| | 1.9 | 1.5 |
| Mean | 1.3 ± 0.5 | 1.4 ± 0.2 |
| *Mytilus edulis* | | |
| | 1.4 | 0.9 |
| | 1.4 | 1.2 |
| Mean | 1.4 ± 0.0 | 1.0 ± 0.1 |
| Bahia Bustamante | | |
| *Aulacomya atra* | | |
| | 1.3 | 1.3 |
| | 1.8 | 1.2 |
| | 1.8 | 1.4 |
| | 1.8 | 1.2 |
| Mean | 1.7 ± 0.2 | 1.3 ± 0.1 |
| *Mytilus edulis* | | |
| | 1.7 | 0.9 |
| | 1.7 | 1.0 |
| | 1.6 | 1.0 |
| | 1.5 | 0.8 |
| Mean | 1.6 ± 0.1 | 0.9 ± 0.1 |
| Bahia Bustamante South | | |
| *Aulacomya atra* | | |
| | 1.8 | 1.3 |
| | 1.8 | 1.6 |
| | 1.4 | 1.4 |
| | 1.6 | 1.6 |
| | 1.7 | 1.6 |
| Mean | 1.7 ± 0.1 | 1.5 ± 0.2 |

**Table 2.** *Cont.*

| Locality | δ¹³C‰ (V-PDB) | δ¹⁸O‰ (V-PDB) |
|---|---|---|
| Bahia Solano *Aulacomya atra* | | |
| | 1.7 | 1.5 |
| | 1.6 | 1.1 |
| Mean | 1.7 ± 0.0 | 1.3 ± 0.2 |
| *Mytilus edulis* | | |
| | 1.6 | 1.7 |
| | 1.4 | 1.0 |
| | 1.7 | 1.0 |
| Mean | 1.6 ± 0.1 | 1.2 ± 0.4 |
| Rada Tilly *Aulacomya atra* | | |
| | 2.2 | 1.4 |
| | 2.2 | 1.2 |
| | 1.7 | 1.5 |
| | 2.3 | 1.5 |
| Mean | 2.1 ± 0.3 | 1.4 ± 0.1 |
| *Mytilus edulis* | | |
| | 1.8 | 1.2 |
| | 1.3 | 1.0 |
| | 1.3 | 1.2 |
| Mean | 1.5 ± 0.3 | 1.1 ± 0.1 |
| Caleta Olivia *Aulacomya atra* | | |
| | 1.5 | 1.5 |
| | 1.7 | 1.5 |
| | 1.9 | 1.7 |
| | 1.9 | 1.7 |
| | 1.5 | 1.5 |
| | 1.6 | 1.4 |
| Mean | 1.6 ± 0.2 | 1.6 ± 0.1 |

One group of fossil samples belongs to Mytilidae *Brachidontes purpuratus* and two groups of samples belong to *Nacella (Patinigera) deaurata* (Table 3).

**Table 3.** Isotopic composition (oxygen and carbon) from samples from Holocene sections along the Camarones coastal area. For location see Figure 1c.

| Species | δ¹³C‰ (V-PDB) | δ¹⁸O‰ (V-PDB) | Age (a BP) |
|---|---|---|---|
| *Aulacomya atra* | 2.2 | 1.4 | 915 ± 20 |
| | 2.7 | 1.6 | |
| | 2.3 | 1.2 | |
| | 2.3 | 1.3 | |
| | 2.1 | 1.0 | |
| | 1.7 | 1.4 | |
| | 2.3 | 1.4 | |
| | 1.5 | 1.4 | |
| | 2.0 | 1.6 | |
| Mean | 2.1 ± 0.4 | 1.4 ± 0.2 | |

**Table 3.** *Cont.*

| Species | δ¹³C‰ (V-PDB) | δ¹⁸O‰ (V-PDB) | Age (a BP) |
|---|---|---|---|
| *Brachidontes purpuratus* | 2.5 | 1.3 | |
| *Aulacomya atra* | 1.8 | 1.1 | 4074 ± 50 |
| | 1.6 | 1.1 | |
| | 1.6 | 1.2 | |
| | 1.2 | 1.1 | |
| Mean | 1.6 ± 0.3 | 1.2 ± 0.02 | |
| *Patinigera deaurata* | 1.1 | 1.4 | |
| | 1.2 | 1.2 | |
| Mean | 1.2 ± 0.0 | 1.3 ± 0.1 | |
| *Mytilus edulis* | 1.7 | 1.3 | 5132 ± 67 |
| | 1.7 | 1.3 | |
| Mean | 1.7 ± 0.0 | 1.3 ± 0.15 | |
| *Mytilus edulis* | 1.9 | 1.2 | 5562 ± 43 |
| | 1.9 | 1.22 | |
| Mean | 1.9 ± 0.0 | 1.2 ± 0.0 | |
| *Mytilus edulis* | 2.6 | 1.3 | 5675 ± 45 |
| | 2.8 | 1.5 | |
| | 2.4 | 1.1 | |
| | 2.8 | 1.2 | |
| Mean | 2.6 ± 0.2 | 1.3 ± 0.2 | |
| *Mytilus edulis* | 2.1 | 2.5 | 6350 ± 20 |
| | 2.1 | 2.4 | |
| | 2.7 | 2.5 | |
| | 2.2 | 2.2 | |
| | 1.8 | 2.2 | |
| | 2.2 | 2.5 | |
| | 2.0 | 2.2 | |
| | 2.5 | 2.7 | |
| Mean | 2.2 ± 0.3 | 2.4 ± 0.2 | |
| *Mytilus edulis* | 1.8 | 2.4 | 6486 ± 46 |
| | 1.9 | 2.3 | |
| | 1.7 | 2.5 | |
| | 2.1 | 2.2 | |
| Mean | 1.9 ± 0.2 | 2.3 ± 0.2 | |

Few superficial marine waters were collected during the 2011 and 2010 fieldwork (Table 4). The δ¹⁸O values were determined after equilibration with $CO_2$ [42] and measured with a Thermo Finnigan MAT-252 mass spectrometer. The data are reported in per mille values (‰) with respect to V-SMOW. Analytical reproducibility was usually 0.1‰.

**Table 4.** Oxygen isotope composition, temperature, salinity and pH of seawater collected along the Camarones coast.

| Lat. | Long. | Date of Sampling | Conductibility (25 °C mS) | pH | Cl (ppm) | T (°C) | δ¹⁸O V-SMOW (‰) |
|---|---|---|---|---|---|---|---|
| S44°42′51″ | O065°40′31.80″ | 2/2011 | 44.44 | 8.04 | - | 18.5 | −1.07 |
| S44°42′51″ | O065°40′31.80″ | 2/2010 | - | - | 18,710 | - | −0.66 |
| S44°50′19.5″ | O065°43″05.1″ | 2/2011 | 48.58 | 8.10 | 18,923 | 21.5 | −0.38 |
| S44°52′54.7″ | O065°40′12.0″ | 2/2011 | 45.51 | 8.04 | 18,887 | 18.5 | 0.03 |
| S45°04″23.6″ | O066°26′46.8″ | 2/2011 | 42.36 | 8.02 | 19,030 | 17.5 | −0.40 |
| S44°59′12.1″ | O066°11′50.9″ | 2/2011 | 43.45 | 8.03 | 18,852 | 15.6 | −0.41 |

*Mytilus edulis* (the 'blue mussel') is an epifaunal suspension feeder found in intertidal-up-to-subtidal environments (reported to be up to 25 m in depth) attached to rocks and to other hard substrates [43], or as loose beds on sandy substrata [44]. The mussel is eurythermal and well acclimated in a +5 to +20 °C temperature range, with an upper sustained thermal tolerance limit of about +29 °C for adults [45]. It does not thrive in salinities lower than 15‰. *Aulacomya atra* is an epifaunal suspension feeder species living in intertidal-up-to-subtidal environments generally as low as 30/40 m [46], and attached to rocks and to other hard substrates. It is euryhaline, able to withstand values higher or lower than those of typically marine species. It is, however, also reported to adapt to intertidal pools with low salinity (15–20‰).

*Brachidontes purpuratus* is an epibyssate and a filter feeder species living on hard bottoms in shallow littoral areas (supratidal to infralittoral zones). This species has been recorded in intertidal highly energetic marine waters among algae or in estuarine zones in low-salinity pools [10]. The species *Nacella (Patinigera) deaurata* is an epibenthic, sessile grazing feeder, found on the rocky shores of supralittoral, intertidal and sublittoral environments up to 30 m deep [47,48]. No data have been found on the salinity tolerance of this species. However, as it is also an intertidal form, it is certainly euryhaline with an ability to withstand values different from those that are typically marine.

Mytilidae shells are a mix of aragonite and calcite layers [49]. As aragonite can turn into calcite during the processes of alteration and diagenesis, a set of XRD analyses were performed on living and fossil shells to confirm the existence of the mixture.

Owing to the wide variability of isotopic composition resulting from the process of shell accumulation, we normalised the isotopic composition to 100% of calcite, without considering the ratio of calcite/aragonite, as performed for instance by [18]. Indeed, the identified isotopic variability is within the potential correction because of the different isotopic fractionation of the calcium carbonate polymorphs in both carbon and oxygen [50–53].

## 4. Results

Both modern and Holocene shells are well preserved (usually maintaining their original colour). XRD analyses show that the Holocene shells are a mixture of aragonite and calcite, just like the modern specimens, confirming that they have not experienced diagenetic transformation. Modern samples collected at different sites show a relatively narrow range of isotopic variability (<1‰) for both the carbon and the oxygen isotope ratio (Table 2). The isotopic variability of the modern populations can be accounted for the fact that Mytilidae usually live for only a few years, when they experience different environmental conditions (i.e., salinity and temperature) in different seasons and different years, also related to the local tidal conditions. This finding is consistent with the observation that in living populations, different individuals show small differences and they offset from the isotopic equilibrium condition [54]. Furthermore, the collected shells are the result of storm accumulation on the beach; shells of different ontogenetic ages (even if only adult individuals were selected) and living on different microenvironments (different depths, tidal conditions, etc.) are mixed together. This variability is usually replicated in the fossils, which were accumulated under the same processes (Figure 2B), even if the isotopic range is usually higher. The selection of shells with joined valves at the same level significantly reduces the possibility of having collected reworked individuals related to different accumulation events and/or, in the worst of cases, eroded from older layers. However, a wider variability can be expected in fossil accumulation than in living shells usually collected in a single storm berm. Fossil accumulation can be the result of the more complex accumulation of several storms and possible later stages of further shell dispersion.

All in all, modern shells of *M. edulis* and *A. atra* collected in the same storm accumulation show very small differences in isotopic composition, substantially overlapping in terms of oxygen and carbon isotope composition (Table 5). This indicates that the $\delta^{18}O$ and $\delta^{13}C$ values of *M. edulis* and *A. atra* in the area can be used interchangeably, with negligible error, for the reconstruction of past coastal conditions.

**Table 5.** Average isotopic differences between *Mytilus edulis* and *Aulacomya atra* in different localities of S. Jorge Gulf.

| Locality | $\delta^{18}O_{Mytilus}-\delta^{18}O_{Aulacomya}$ (‰) | $\delta^{13}C_{Mytilus}-\delta^{13}C_{Aulacomya}$ (‰) |
|---|---|---|
| *Camarones South* | −0.31 | 0.07 |
| *Bahia Bustamante* | −0.38 | −0.05 |
| *Bahia Solano* | 0.03 | 0.35 |
| *Rada Tilly* | −0.27 | −0.65 |
| Mean | −0.23 ± 0.18 | −0.07 ± 0.42 |

Shell isotopic data of the Holocene sections (Figures 3 and 4) younger than ca. 6100 a cal BP, including living shells, overlap along a relatively narrow range of $\delta^{18}O$ values (between ca. +1.50 and +1.00‰). Older samples show significantly higher $\delta^{18}O$ values ranging from ca. +2 to +2.5‰, with no overlaps with the younger samples. Carbon isotope compositions show a more complex pattern of variability, with partial overlaps for samples of different ages ranging from ca. +1.1 to +2.8 ‰ (Figures 3 and 4).

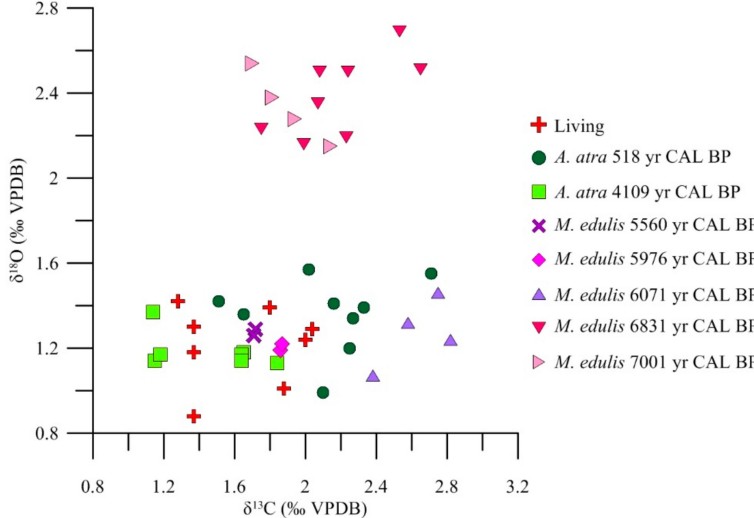

**Figure 3.** Oxygen and carbon isotope composition for fossil shells. Median calibrated radiocarbon age is reported for each group.

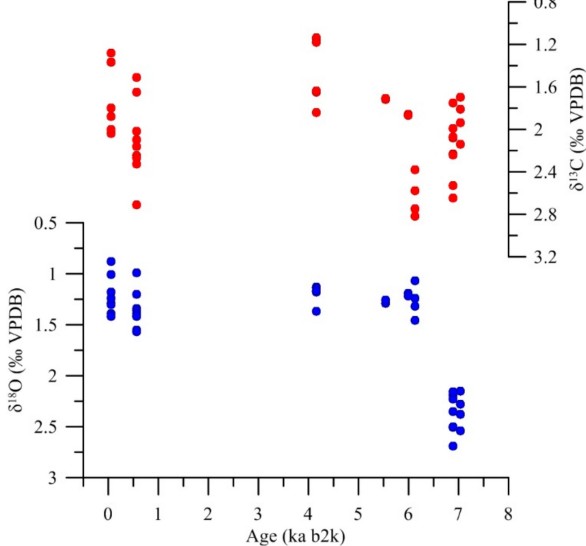

**Figure 4.** Oxygen and carbon isotope composition for fossil shells. Median calibrated radiocarbon age for each group is reported.

## 5. Discussion

The $\delta^{18}O$ values of the mollusc shells are functions of both $\delta^{18}O$ of seawater, temperature [51,55–57], and additional 'vital effects' [56,58,59]. Wanamaker et al. [60] proposed the following empirical relation (Equation (1)) between temperature and isotopic composition of water and of carbonate shells for *M. edulis*:

$$T°C = 16.28(\pm 0.10) - 4.57(\pm 0.15)\,[\delta^{18}O_c - \delta^{18}O_w] + 0.06\,[\delta^{18}O_c - \delta^{18}O_w]^2 \qquad (1)$$

where $\delta^{18}O_w$ is the isotopic composition of water with respect to V-SMOW, while $\delta^{18}O_c$ is the isotopic composition of carbonate with respect to V-PDB. This implies a $\Delta\delta^{18}O_c/°C$ of ca. $-0.2‰/°C$ between ca. 5 and 25 °C. This equation agrees with the experimental data [53] obtained in precipitated carbonate close to isotopic equilibrium (however, for isotopic equilibrium on marine carbonate, see [61]).

At glacial to interglacial scale, global seawater $\delta^{18}O_w$ values are mostly related to changes in the amount of water stored in continental ice [62,63]. A change of ca. 0.009‰/m in the eustatic sea-level component is estimated globally [63,64], with lower sea-level characterised by higher $\delta^{18}O$ seawater values. Surface seawater isotopic composition is also influenced by changes in the evaporation and/or dilution with freshwater, which is often marked by changes in salinity [65,66], characterising different oceanic currents.

According to [13], the $\delta^{18}O$ seawater value in the study area today should be around $-0.3$–$-0.2‰$. This value is slightly higher than the few isotopic data obtained on the samples occasionally collected in the area (Table 4), which yielded a $\delta^{18}O$ average value of $-0.48 \pm 0.36‰$. We obtained an average calcification temperature of ca. 11.3 °C (Equation (1)) by using the average $\delta^{18}O$ values of the modern shells collected in the Camarones area ($1.03 \pm 0.15‰$ for *M. edulis*, Table 2) and the $\delta^{18}O$ values of seawater of $-0.2‰$ (Equation (1)), close to local average temperature; instead, we obtained a value of ca. 9.5 °C by using the measured isotopic values of seawater in Table 4. The measured $\delta^{18}O_w$ values underestimate the temperature probably as a result of the changes in seasonal and local salinity.

Colonese et al. [21] and Yan et al. [67] proposed to evaluate the regional $\delta^{18}O_w$ by using a relation with sea surface salinity (SSS) (Equation (2)):

$$\delta^{18}O_w\,(‰) = 0.30\,SSS - 10.52 \qquad (2)$$

The $\delta^{18}O_w$ values of seawater will range from ca. $-0.6$ to $-0.5‰$ by introducing the average SSS values reported for the area by [37]. This suggests that Equation (2) is not an excellent predictor for the isotopic composition of seawater in the area, even if the obtained results are still reasonable. This brief discussion, however, seems to confirm that by using a correct value of $\delta^{18}O_w$ seawater, Equation (1) correctly reconstructs the annual average temperature, so that all the *M. edulis* shells could be used locally for this purpose.

The carbon isotope composition of mollusc shells depends mostly on dissolved inorganic carbon (DIC) isotopic composition and the metabolic $CO_2$ amount involved in shell calcification [59,68,69]. Significant offsets compared to isotopic equilibrium conditions are reported for different species, and these offsets are generally species-dependent [70–72]. The carbon isotope composition of *M. edulis* has been used in the past for tracking changes in DIC isotopic composition and salinity related to mixing processes between different water masses [60,73]. For *M. edulis*, Wanamaker et al. [60] found that $\delta^{13}C$ values were substantially lower than predicted values at equilibrium [52] and that the differences between the $\delta^{13}C$ measured and the $\delta^{13}C$ at the isotopic equilibrium increased with increasing salinity. According to [74], the incorporation of different amounts of metabolic $CO_2$ in *M. edulis* can reduce the use of $\delta^{13}C$ as an indicator of DIC and salinity, but may be useful to assess metabolic differences between different populations. However, it can be used as an indicator of large $\delta^{13}C$ changes in the isotopic composition of DIC and of salinity. In our case, the use of $\delta^{13}C$ data was further complicated by the use of other Mytilidae species for which experimental data are not available (see Table 5). However, no great differences seem to be present in the whole set of data. Therefore, no dramatic changes seem

to have occurred in the DIC of the area, which may also support relatively minor changes in seawater salinity, with no substantial mix with fresher or saltier waters.

However, the most relevant and interesting features of the isotopic data regard the high $\delta^{18}$O values shown by samples older than ca. 6100 cal yr BP (Figure 4). Before ca. 6100 cal a BP, the average $\delta^{18}$O value is +2.4 ± 0.2‰, whereas the average value for younger samples (including the modern ones) is +1.3 ± 0.2‰. According to Equation (1), this corresponds to a lower temperature (ca. 5 °C) for the samples older than 6100 a cal BP if no change has occurred in the isotopic composition of seawater. The eustatic sea-level component for the last ca. 7000 a shows no significant variations that could justify great changes in seawater isotopic composition [64]. However, an estimation of ca. 5 °C cooler water is probably a maximum value, without considering any changes in salinity. However, it is of particular interest to note that, associated with the larger changes in the $\delta^{18}$O values at ca. 6.1 cal ka BP, there are no equivalent changes in the $\delta^{13}$C of the mollusc shells, suggesting that the no particular changes in seawater condition (i.e., salinity and DIC) occurred.

This estimated temperature value may contrast with paleontological evidence, which suggest ca. 1–3 °C warmer waters for this part of Patagonia during the middle Holocene [75]. However, the period of warmer temperatures inferred from mollusc analyses is not chronologically well-constrained and it may have occurred in some unsampled periods because of the discontinuity in our chronological records. For instance, model simulations tend to place the Holocene thermal optimum at high latitudes in the Southern hemisphere between ca. 6000 and 3000 cal a BP [73]. However, our isotopic data do not support specific warmer intervals compared to the present conditions for the investigated record, if the data are simply interpreted in terms of temperature changes.

On the other hand, the higher isotopic composition recorded in the period before ca. 6.1 cal ka BP might have been due to changes in salinity (i.e., more evaporated waters). By assuming no significant changes in surface temperature compared to the present conditions, we can calculate seawater $\delta^{18}$O values of ca. +1.4‰ (Equation (1)), which appear unrealistic for the present-day distribution of isotopic seawater composition [13], and these values should be accompanied by more unrealistic changes in salinity (ca. 48‰, from Equation (2)). Therefore, a plausible explanation is that most of the changes in the measured $\delta^{18}$O values were driven by temperature, even if we cannot rule out a minor change in local salinity, which is not completely supported by carbon isotopic composition.

Considering that temperature could be the main component, the only reliable explanation for such great mutations in temperature is a change in the energy and influence of the Falkland Current in the area. According to [4], the shift in the position of the Brazilian and Malvinas currents confluence occurred several times during the Holocene, as indicated by stable isotopes in the *G. inflata* of the marine GeoB13862-1 core (Figure 5C).

In particular, before ca. 6 cal ka BP, there was a tendency toward a northward position of the confluence zone. This may have produced a major and stronger influence of the Falkland current in the Camarones area, which rendered the seawater cooler than in younger periods. Warmer sea conditions might have occurred later during a period of maximum southward shift in the confluence zone between 4.5 and 5.5 cal ka BP (Figure 5C), thus favouring the penetration and survival of warmer marine mollusc association, as indicated by paleontological studies [75]. The northward position of the confluence before ca. 6.1 cal ka BP might have decreased the temperature with no major changes in terms of isotopic sea-water composition compared to the present-day conditions. According to [13], local seawater oxygen isotopic composition does not show an important isotopic gradient over the influence of the Falkland current, as also observable in marine core data [4]. Voigt et al. [4] interpreted the change in the position of current confluence as a consequence of the change in the position and strength of the South Westerly Winds. In this perspective, it is worth considering the correlation with the oxygen isotope composition of carbonates from the Laguna Potrok Aike ([73], Figure 5B) in southern Patagonia. This proxy is interpreted as an indication of the strength and position of the southern westerlies. According to [73], progressive warming after the Last Glacial Maximum produced weak Westerlies during the Lateglacial and early Holocene, interrupted by an interval with

strengthened Westerlies between ca. 13.4 and 11.3 cal ka BP. Wind strength increased at 9.2 cal ka BP and significantly intensified until 7.0 cal ka BP. Subsequently, wind intensity diminished and stabilised to conditions similar to those of the present day (Figure 5B).

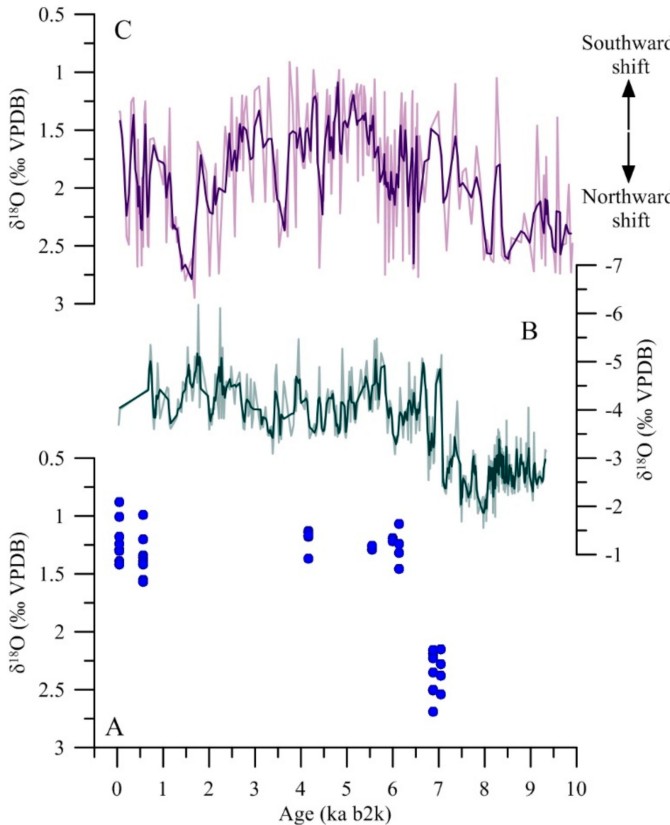

**Figure 5.** (**A**) Oxygen isotope records from this work plotted against isotopic composition of *G. inflata* [4] in the Brazilian and Falkland confluence zone (**C**) and oxygen isotopic composition of carbonates from laguna Potro Aike (**B**) [73].

If our interpretation is correct, the oxygen isotope data obtained for the Mytilidae shells collected from the Camarones area support the hypothesis according to which the intensity changes in the Falkland Current during the Holocene may have been driven by the changes in the position of the Westerlies (Figure 5). The continental and marine proxies selected and shown in Figure 5 [4,73] are consistent with this interpretation.

According to [76], several proxy records of southernmost Patagonia show an Early Holocene thermal maximum between ca. 11.5 and 8.5 ka and lower temperatures during the last 8 ka. For this sector of the Atlantic coast of Patagonia, our data show a warming after ca. 6 cal ka BP, suggesting that in this sector, the effect of the Falkland current delayed the occurrence of a 'thermal maximum' of some millennia.

## 6. Conclusions

The oxygen isotope composition of the Mytilidae collected near the Camarones village shows that important changes occurred between ca. 6800 and 6100 cal a BP. Analysis of the data seems to suggest that this condition may have been related to important changes in temperature, probably at a maximum of ca. 5 °C. This change is likely to have been caused by a northward shift in the position of the confluence of the Falkland and Brazilian currents, driven by a change in the strength and position of the Westerlies. According to the isotopic data presented in our paper, the presence of mollusc association, indicating warmer conditions (1–3 °C) during the middle Holocene, was probably younger than ca. 6800 cal a BP. However, the low resolution achieved in this pilot study and the poor chronological

constraint for mollusk association do not allow us to be more precise. However, these data indicate that raised marine terraces over the Patagonian coast may contain valuable paleoceanographic information, which can be improved in resolution for the Holocene and extended back to the older Interglacial.

**Author Contributions:** Conceptualization, G.B., I.C. and G.Z.; methodology, I.C., I.B., G.B. and G.Z.; validation, I.B., M.G., I.I., F.T. and L.D.; formal analysis, I.C., I.B., M.G., E.R., F.T. and L.D.; investigation (including field work), I.C., G.B., G.Z., I.B., I.I., M.B. and L.R.; resources, G.Z., M.G. and L.D.; data management, I.C., I.I., M.B. and G.B.; original draft preparation, I.C., G.B. and G.Z.; supervision, G.Z.; project administration, G.Z.; funding acquisition, G.Z. All authors have read and agreed to the published version of the manuscript.

**Funding:** This work was funded by the University of Pisa (Progetto Ateneo 2007, Leader G. Zanchetta; Progetto Ateneo PRA 2015 Leader G. Zanchetta) and MIUR (PRIN2008, Leader G. Zanchetta

**Acknowledgments:** We wish to thank J. Cause and the no-profit CADACE organization for the logistical support provided in the field campaign.

**Conflicts of Interest:** The authors declare no conflict of interest.

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
