# Peer review of "Stable Oxygen and Carbon Isotope Composition of Holocene Mytilidae from the Camarones Coast (Chubut, Argentina): Palaeoceanographic Implications"

_water, doi:10.3390/w12123464_

Round 1
Reviewer 1 Report
Dear Authors,
The collected data are interesting and the relationship with the regional current fluctuations after the Younger Dryas worth to be published.
Some aspects should be more carefully considered:
It is not clear which of the 14C data from Table 1 were already published in 2017-2018 Holocene paper.
Figure 1 and 2 are of low quality. They should be merged and displayed as in the Holocene paper, vertically. Some other suggestions are insert directly in the manuscript. The position of the samples and the mentioned currents should be shown. In the present form the position of samples are poorly displayed in Fig. 2, hard to read and follow.
In Discussion a paragraph concerning the relationship with progressive warming and fluctuations after the Younger Dryas (beginning of Holocene) should be insert.
Other suggestion are find directly insert on the manuscript.

Author Response
Dear Editor,
We thank the five reviewers for their positive comments and useful suggestions. We have virtually accepted almost all their suggestions and a detailed reply is reported below. Our replies are in italics, and all the corrections are reported in red in the main text.
Reviewer 1
-The collected data are interesting and the relationship with the regional current fluctuations after the Younger Dryas worth to be published.
We thank for the positive comment of ref#1 and for the very careful review.
-Some aspects should be more carefully considered: It is not clear which of the 14C data from Table 1 were already published in 2017-2018 Holocene paper.
There are no radiocarbon ages from the paper published in Bini et al., 2018. . Only three ages come from Zanchetta et al., 2012. This is now clearly stated in the caption of Table 1.
-Figure 1 and 2 are of low quality. They should be merged and displayed as in the Holocene paper, vertically. Some other suggestions are insert directly in the manuscript. The position of the samples and the mentioned currents should be shown. In the present form the position of samples are poorly displayed in Fig. 2, hard to read and follow.
We have merged Figs. 1 and 2 and improved the quality of the original Fig. 2, where the position of the samples were reported correctly. All the corrections requested (see below) have been performed.
-In Discussion a paragraph concerning the relationship with progressive warming and fluctuations after the Younger Dryas (beginning of Holocene) should be insert.
This is not specifically the target of our paper (our records are shorter than those of the YD). However, we have inserted some sentences on that point, as indicated in the attached pdf from reviewer 1.
-Other suggestion are find directly insert on the manuscript.
Please consider the pdf sent by ref#1
Figure 1 Please insert the position of the two currents mentioned, namely the Falkland and Brazilian currents. A position for the pre-7000 years and younger than 6000 years would help the reader to better understand the scenario.
The position of the two currents is now reported in Fig. 1. However, in our opinion it is not possible to reconstruct the position of the confluent zone in different periods (at least for the data available), without inserting a very speculative reconstruction.
Line 89 elevated soil evapotranspiration It would be interesting to insert some numbers instead of "elevated".
This section has been expanded (lines 92-99):
“The meteorological station of the Camarones village records a mean annual precipitation of 287 mm/a with a mean annual temperature of 12.6 °C. Throughout the year, the strong activity of the southern westerlies causes elevated soil evapotranspiration, turning the area into a semiarid and poorly vegetated land [34]. The potential evapotranspiration, calculated by the Thornthwaite method, indicates an annual value of <700 mm, with a maximum of ≥100 mm in January (warm season) and a minimum of 15 mm in June (cold season). The annual evapotranspiration exceeds the rainfall, so that the annual water deficit reaches approximately 450 mm, without considering the additional effect of the strong winds (http://www.ambiente.chubut.gov.ar).”
Line 93 Range of variation for the whole year?
This section has now been expanded (lines 103-108):
“For this area, Falabella et al. [37] and the Argentine Naval Hydrography Service (http://www.hidro.gov.ar/ceado/Ef/Inventar.asp) reported an annual mean sea-surface temperature (SST) of around 12 °C; an average SST of 15.6°C in the warm season; an average of 7.7°C in the cold season and a sea surface salinity (SSS) between ca. 33 and 33.5 ‰. This is in agreement with the satellite data, which indicate an average temperature of the sea off Camarones of 12 ± 3 °C for 2015 (http://seatemperature.org).”
-Line 104 in order.
Line 104 has been changed, and is now line 116
-Table 1 Please mention which data were firstly published in Bini, M.; Isola, I.; Zanchetta, G.; Pappalardo, M.; Ribolini, A.; Ragaini, L.; Baroni, C.; Boretto, G.; 429 Fuck, E.; Morigi, C.; Salvatore, M.C.; Bassi, D.; Marzaioli, F.; Terrasi, F. 2018. Middle Holocene relative 430 sea-level along Atlantic Patagonia: new data from Camarones (Chubut, Argentina). The Holocene, 431 2018, 28, 56–64.
We have replied to this above. No radiocarbon data have been used by Bini et al. 2018. These dated layers were not used for stable isotopes. In our manuscript, we have only quoted the layers for which there was sufficient shell material for isotope analyses. The information in the caption of Table 1 is now complete.
-Table 1 It is not clear which samples are coming from where. Please add on Fig 2 more clearly this.
The position of the samples was originally reported in Fig. 2. This figure has now been improved and the information has been inserted in Fig. 1.
-Also Fig 1 and 2 should be vertical and not horizontal. In fact Fig. 1 and 2 can be merged to one figure with better graphic and description. Current position should be also insert.
Figures 1 and 2 have been merged. The position of the currents is now reported.
-Tables As the standard deviation is almost 0.2 permil (in any case larger than 1 permil) the second decimal for the isotopic values is not representative. Should be removed in text and tables and the numbers rounded to one decimal.
Yes, we perfectly agree. This has been done throughout the text
-Line 241 Deleted shells
Done (now line 257)
-Line 241 changes with” lower”
Done (now line 257)
-Line 242.
The sentence has been changed as follows:
“…that the differences between the δ13C measured and the δ13C at the isotopic equilibrium increased with increasing salinity.” (now line 258)
-Line 250 commas:
This sentence has been deleted.
-Line A paragraph concerning the relationship of this event with progressive warming and fluctuations after the Younger Dryas event (beginning of Holocene) should be insert.
We have inserted the following sentences at lines (321-325)
“According to [78], progressive warming after the Last Glacial Maximum produced weak Westerlies during the Lateglacial and early Holocene, interrupted by an interval with strengthened Westerlies between ca. 13.4 and 11.3 cal ka BP. Wind strength increased at 9.2 cal ka BP and significantly intensified until 7.0 cal ka BP. Subsequently, wind intensity diminished and stabilized to conditions similar to those of the present day (Fig. 6B).”
-Fig. 5 Why no populations in the interval 0.5 to 4 ka? Some plausible explanation?
In the area there are evidences of depositional phases partially covering this interval (e.g. Shellmann and Radke, 2010), but there are no shell accumulations useful to our purposes.
-Line 6 Conclusion Shift references from Conclusions to Discussion. In Conclusions only conclusions regarding the present work should be insert.
The conclusions have been changed as follows:
“The oxygen isotope composition of the Mytilidae collected near the Camarones village shows that important changes occurred between ca. 6800 and 6100 cal a BP. Analysis of the data seems to suggest that this condition may have been related to important changes in temperature, probably at a maximum of ca. 5°C. This change is likely to have been caused by a northward shift in the position of the confluence of the Falkland and Brazilian currents, driven by a change in the strength and position of the Westerlies. According to the isotopic data presented in our paper, the presence of mollusc association, indicating warmer conditions (1-3°C) during the middle Holocene, was probably younger than ca. 6800 cal a BP. However, the low resolution achieved in this pilot study and the poor chronological constraint for mollusk association do not allow us to be more precise. However, these data indicate that raised marine terraces over the Patagonian coast may contain valuable paleoceanographic information, which can be improved in resolution for the Holocene and extended back to the older Interglacial.”
-Line 318 deleted “for samples”.
The sentence is no longer present.
Reviewer 2 Report
The manuscript of Boretto et al. deals with a very interesting question, paleoclimatic and paleoceanographic interpretation of stable isotope compositions of marine bivalves from the Patagonian coast. The paper is well written, concise, clear, all the interpretations are supported by the data and independent evidences. Language and style are fine. The conclusion of changing strengths of competing ocean currents has a wider implication, as it can explain why the Holocene Climate Optimum appears „late” in the Patagonian sequences. The results logically suggest that the the HCO is missing there due to the cooling affect of the Falkland current. So it is not the HCO arriving late, but actually its effect is masked by the cool current from the south. Without HCO the local temperature would have been even lower.
There are really minor corrections and changes that I suggest, so I propose acceptance with minor revision.
Specific comments:
- Line 22: Analysis.
- The word „confluence” is used three times in two sentences in the Abstract. I suggest rephrasing.
- Line 103: some explanation is needed why H2O2 was used instead of HclO, although H2O2 treatment can affect the carbonate’s d18O value.
- Tables 2 and 3: it is not necessary to list all the measurements, just keep the averages and scatters.
- Line 162: actually the samples are normalized as 100% calcite as they were measured against calcite standards. It would be useful to make a calculation of the aragonite correction effect with the given aragonite % values (XRD can give a hint on it).
- Line 196: Carbon isotope compositions show...
- Line 250: The difference is so small that measuring more samples may diminish this temporal δ13C change. This should not be interpreted, just say that no real δ13C change is visible. That’s fine for the later discussion on salinities.
- Line 274: the wider implication on the appearance or missing of Holocene Climate Optimum could be discussed here in 1-2 sentences.
Author Response
Dear Editor,
We thank the five reviewers for their positive comments and useful suggestions. We have virtually accepted almost all their suggestions and a detailed reply is reported below. Our replies are in italics, and all the corrections are reported in red in the main text.
Reviewer 2
The manuscript of Boretto et al. deals with a very interesting question, paleoclimatic and paleoceanographic interpretation of stable isotope compositions of marine bivalves from the Patagonian coast. The paper is well written, concise, clear, all the interpretations are supported by the data and independent evidences. Language and style are fine. The conclusion of changing strengths of competing ocean currents has a wider implication, as it can explain why the Holocene Climate Optimum appears „late” in the Patagonian sequences. The results logically suggest that the the HCO is missing there due to the cooling affect of the Falkland current. So it is not the HCO arriving late, but actually its effect is masked by the cool current from the south. Without HCO the local temperature would have been even lower.
There are really minor corrections and changes that I suggest, so I propose acceptance with minor revision.
Specific comments:
- Line 22: Analysis.
Thank you. The correction has been made (now line 25)
- The word „confluence” is used three times in two sentences in the Abstract. I suggest rephrasing.
Thank you. The sentence has been rephrased (lines 28-31).
“A possible explanation of this change can be related to a northward position of the confluence zone of the Falkland and Brazilian currents. This is consistent with the data obtained in marine cores, which indicate a northerly position of the confluence in the first half of the Holocene.”
- Line 103: some explanation is needed why H2O2 was used instead of HclO, although H2O2 treatment can affect the carbonate’s d18O value.
It is well known that a soft leaching using HCl is used for preparing samples for radiocarbon dating to dissolve any secondary carbonate on the surface of the sample. This pretreatment is not used to oxidize organic matter and would be too strong for this action for samples treated for stable isotope analyses.
Moreover, most of the discussion on the isotopic effect during pretreatment of carbonate for oxidizing the organic matter is usually related to powdered samples, where the specific surface between solution and samples is huge favoring possible isotopic exchange. The application of H2O2, to whole samples has no basically effect. However, recent modern researches on the isotopic effect during pretreatment show small and unpredictable effect and for almost pure carbonate no “hard” pretreatment should be preferred (Mannella et al., 2020).
This is well reported in the text:
“Whole shells were immersed for ca 24 in a H2O2 solution so as to remove remains of organic matter. They were then washed in ultrasonic bath and rinsed several times with deionised water. One shell of the pair was grounded in an agate mortar and was used for stable isotope analyses. A similar procedure was followed for fossil shells selected from Holocene deposits (Fig. 2). The powder obtained was not subject to further pre-treatments to avoid undesirable isotopic effects.”
- Tables 2 and 3: it is not necessary to list all the measurements, just keep the averages and scatters.
This is probably right, but it is useful to have all the data in the tables.
- Line 162: actually the samples are normalized as 100% calcite as they were measured against calcite standards. It would be useful to make a calculation of the aragonite correction effect with the given aragonite % values (XRD can give a hint on it).
We have changed the text accordingly (lines 176-178).
“Owing to the wide variability of isotopic composition resulting from the process of shell accumulation, we normalized the isotopic composition to 100% of calcite, without considering the ratio of calcite/aragonite, as performed for instance by [18].”
- Line 196: Carbon isotope compositions show...
Thank you, this line has been corrected (it is now line 211)
- Line 250: The difference is so small that measuring more samples may diminish this temporal δ13C change. This should not be interpreted, just say that no real δ13C change is visible. That’s fine for the later discussion on salinities.
Yes, we agree. Changes have been made, and the section is now as follows (lines 264-266):
“However, no great differences seem to be present in the whole set of data. Therefore, no dramatic changes seem to have occurred in the DIC of the area, which may also support relatively minor changes in seawater salinity, with no substantial mix with fresher or saltier waters.”
- Line 274: the wider implication on the appearance or missing of Holocene Climate Optimum could be discussed here in 1-2 sentences.
Yes, a couple of sentences have been added at the end of the discussion (Lines 334-338)
“According to [79], several proxy records of southernmost Patagonia show an Early Holocene thermal maximum between ca. 11.5 and 8.5 ka and lower temperatures during the last 8 ka. For this sector of the Atlantic coast of Patagonia, our data show a warming after ca. 6 cal ka BP, suggesting that in this sector the effect of the Falkland current delayed the occurrence of a “thermal maximum” of some millennia.”
Reviewer 3 Report
I am glad to read the MS titled "Stable oxygen and carbon isotope composition of Holocene Mytilidae from the Camarones coast (Chubut, Argentina): palaeoceanographic implications" by Boretto. The data is very interested and worthy to be published. But, the Englihsh of the MS should be polished before it is accepted.
Author Response
Dear Editor,
We thank the five reviewers for their positive comments and useful suggestions. We have virtually accepted almost all their suggestions and a detailed reply is reported below. Our replies are in italics, and all the corrections are reported in red in the main text.
Reviewer 3
-I am glad to read the MS titled "Stable oxygen and carbon isotope composition of Holocene Mytilidae from the Camarones coast (Chubut, Argentina): palaeoceanographic implications" by Boretto. The data is very interested and worthy to be published. But, the Englihsh of the MS should be polished before it is accepted.
We thank the reviewer for the positive comments. Along with the comments of the other reviewers and after a careful correction by a native speaker, we have revised and polished the manuscript.
Reviewer 4 Report
The manuscript titled "Stable Oxygen and Carbon Isotope Composition of Holocene Mytilidae from The Camarones Coast (Chubut, Argentina): Palaeoceanographic Implications" is well written, interpretation of the results is consistent.
One of the most important part of research is representativeness of the samples and measuring accuracy. The latter is well described (NBS18 and NBS19 is better to call Reference Materials (as in IAEA page). Also I notice that isotope ratios for the samples are not between these reference materials values, I mean for oxygen NBS19 is -2.2 permill, while samples are in positive delta sites.
Speaking on the experiment design, I would like to read on archive sample collection. Authors cite 30-31 reference, and is not clear, who collected these deposit samples - authors themselves or authors from year 2010-2012? I propose to write in short on archive sample, because it is important.
Another issue is reservoir effect. I understand that if we take this effect on average to few hundred years (MRE can be 510 14C yr (Mangerud and Gulliksen, 1975), 80-1100 yr (Cordero et al 2003). we still can interpret results, spanning to 7000 years back, but then make no sense to write +- 20 years (as Table 2 UCI65213). What is meaning for stars in Table 2?
I propose to clarify in the text and in the Table caption that reservoir effect was not included in the results.
Here we come to the last methodological question - why authors not made 14C measurements for the modern samples? By making 14C in modern samples we can assume the reservoir affect for the site investigated, and use it for data interpretation.
Author Response
Dear Editor,
We thank the five reviewers for their positive comments and useful suggestions. We have virtually accepted almost all their suggestions and a detailed reply is reported below. Our replies are in italics, and all the corrections are reported in red in the main text.
Revisore 4
-The manuscript titled "Stable Oxygen and Carbon Isotope Composition of Holocene Mytilidae from The Camarones Coast (Chubut, Argentina): Palaeoceanographic Implications" is well written, interpretation of the results is consistent.
We thank the reviewer for the positive comments.
-One of the most important part of research is representativeness of the samples and measuring accuracy. The latter is well described (NBS18 and NBS19 is better to call Reference Materials (as in IAEA page).
Thank you, the corrections have been made (line 142).
-Also I notice that isotope ratios for the samples are not between these reference materials values, I mean for oxygen NBS19 is -2.2 permill, while samples are in positive delta sites.
These are usual reference materials for marine carbonates. The internal reference material (MAB1, basically obtained with the same block of marble of IAEA-CO1 and with the same isotopic composition) also has negative (-2.45‰) values. This may have minor negligible effects on our samples.
-Speaking on the experiment design, I would like to read on archive sample collection. Authors cite 30-31 reference, and is not clear, who collected these deposit samples - authors themselves or authors from year 2010-2012? I propose to write in short on archive sample, because it is important.
Thank you for the precious suggestions. Samples were collected by the authors of this paper during field campaigns between 2010 and 2012. The samples are stored in the Laboratory of Paleoclimatology and Geoarcheology of the University of Pisa. The text has been modified accordingly (lines 110-112).
-Another issue is reservoir effect. I understand that if we take this effect on average to few hundred years (MRE can be 510 14C yr (Mangerud and Gulliksen, 1975), 80-1100 yr (Cordero et al 2003). we still can interpret results, spanning to 7000 years back, but then make no sense to write +- 20 years (as Table 2 UCI65213).
Radiocarbon ages are calibrated considering the marine reservoir effect (they were calibrated using the marine curve of Calib13). Our observation was intended to underline that, locally, this may be different from the global marine reservoir proposed by Calib13. To be more clear, we have added a sentence at lines 136-137.
-What is meaning for stars in Table 2?
Sorry, the meaning of the stars is now explained in the caption.
-I propose to clarify in the text and in the Table caption that reservoir effect was not included in the results.
See previous answer. The reservoir corrections have been made.
-Here we come to the last methodological question - why authors not made 14C measurements for the modern samples? By making 14C in modern samples we can assume the reservoir affect for the site investigated, and use it for data interpretation.
Unfortunately this is not so simple. The first age in Table 1 is from a living specimen. However, for the purposes expressed by the reviewer, we should select samples collected before the 1950 AD (i.e. pre-atomic bombs explosions in the atmosphere). We had no possibility to collect these samples which are usually stored (if available) in natural museums. This would require a dedicated approach, which is beyond the aims of this manuscript.
Reviewer 5 Report
The manuscript
Stable oxygen and carbon isotope composition of Holocene Mytilidae from the Camarones coast (Chubut, Argentina): palaeoceanographic implications
by Boretto & coauthors provides novel results after an original investigation in southern Patagonia. The aim of the research is original and interesting, and I appreciate the effort in interpreting these non-obvious results with a very honest approach, which does not minimize the possible bias from insufficient control on historical environmental data.
The paper is well written and paves the way for further interesting discoveries in this underexplored area.
Author Response
Dear Editor,
We thank the five reviewers for their positive comments and useful suggestions. We have virtually accepted almost all their suggestions and a detailed reply is reported below. Our replies are in italics, and all the corrections are reported in red in the main text.
Reviewer 5
-Stable oxygen and carbon isotope composition of Holocene Mytilidae from the Camarones coast (Chubut, Argentina): palaeoceanographic implications by Boretto & coauthors provides novel results after an original investigation in southern Patagonia. The aim of the research is original and interesting, and I appreciate the effort in interpreting these non-obvious results with a very honest approach, which does not minimize the possible bias from insufficient control on historical environmental data. The paper is well written and paves the way for further interesting discoveries in this underexplored area.
We thank the reviewer for the very positive comments
Round 2
Reviewer 1 Report
-